# Validation of a Robotic Testbench for Evaluating Biomechanical Effects of Implant Rotation in Total Knee Arthroplasty on a Cadaveric Specimen

**DOI:** 10.3390/s23177459

**Published:** 2023-08-27

**Authors:** Nikolas Wilhelm, Constantin von Deimling, Sami Haddadin, Claudio Glowalla, Rainer Burgkart

**Affiliations:** 1Department of Orthopedics and Sports Orthopedics, Klinikum rechts der Isar, School of Medicine, 81675 Munich, Germany; 2Munich Institute of Robotics and Machine Intelligence, Department of Electrical and Computer Engineering, Technical University of Munich, 80992 Munich, Germany; 3Department of Trauma and Orthopedic Surgery, Berufsgenossenschaftliche Unfallklinik Murnau, 82418 Murnau, Germany

**Keywords:** knee joint biomechanics, robotic testbench, cadaver study, total knee arthroplasty, implant alignment

## Abstract

In this study, we developed and validated a robotic testbench to investigate the biomechanical compatibility of three total knee arthroplasty (TKA) configurations under different loading conditions, including varus–valgus and internal–external loading across defined flexion angles. The testbench captured force–torque data, position, and quaternion information of the knee joint. A cadaver study was conducted, encompassing a native knee joint assessment and successive TKA testing, featuring femoral component rotations at −5°, 0°, and +5° relative to the transepicondylar axis of the femur. The native knee showed enhanced stability in varus–valgus loading, with the +5° external rotation TKA displaying the smallest deviation, indicating biomechanical compatibility. The robotic testbench consistently demonstrated high precision across all loading conditions. The findings demonstrated that the TKA configuration with a +5° external rotation displayed the minimal mean deviation under internal–external loading, indicating superior joint stability. These results contribute meaningful understanding regarding the influence of different TKA configurations on knee joint biomechanics, potentially influencing surgical planning and implant positioning. We are making the collected dataset available for further biomechanical model development and plan to explore the 6 Degrees of Freedom (DOF) robotic platform for additional biomechanical analysis. This study highlights the versatility and usefulness of the robotic testbench as an instrumental tool for expanding our understanding of knee joint biomechanics.

## 1. Introduction

The human knee, a complex joint essential for movement and weight bearing, requires accurate modeling for a number of applications, such as prosthesis development and human motion analysis [1,2]. The knee joint complex primarily comprises the medial tibiofemoral, lateral tibiofemoral, and patellofemoral articulations. These articulations collectively facilitate six degrees of freedom (DOFs), encompassing rotational motions such as flexion–extension, internal–external, and varus–valgus, as well as translational movements including anterior–posterior, medial–lateral, and compression–distraction [1]. Knee joint biomechanics is an important area of research, as it plays a critical role in understanding the function of the knee, the development of knee pathologies, and the success of surgical interventions such as total knee arthroplasty (TKA) [3,4,5]. In vivo studies have investigated knee joint forces and kinematics after TKA [3,4], while other studies have focused on the role of patient, surgical, and implant design variations in TKA performance [5]. Advanced imaging techniques, such as fluoroscopy and magnetic resonance imaging (MRI), have been utilized to measure natural knee kinematics non-invasively and to conduct in vivo tibiofemoral cartilage-to-cartilage contact area measurements under acute loading conditions [6,7]. Predicting the location of joint centers, such as the hip, is essential for accurate biomechanical modeling and has been addressed in the literature [8]. The development of robotic testing systems for studying knee joint biomechanics can offer valuable insights into the complex interactions between knee joint structures and the effects of various loading conditions, including the impact of implant positioning in TKA. Building upon the need for a comprehensive understanding of knee joint biomechanics, our study is motivated by the significant role knee mechanics play in the initiation and progression of knee osteoarthritis [9,10,11], the influence of knee alignment on disease progression and functional decline [12], and the high complexity and load-bearing requirements of the knee joint that make it prone to injuries and a major cause of disability [13].

Recognizing the necessity for a thorough comprehension of knee joint biomechanics, our study is driven by several key factors. Firstly, knee mechanics significantly influence the onset and progression of knee osteoarthritis [9,10,11]. Secondly, the alignment of the knee has a substantial impact on disease progression and functional decline [12]. Lastly, the knee joint’s inherent complexity and load-bearing demands render it susceptible to injuries, making it a leading cause of disability [13]. Accurate biomechanical testing is crucial for predicting in vivo knee loads [14], developing computational models to analyze knee joint structures [15], and informing surgical planning and implant design for knee arthroplasty [16]. Recent studies have also highlighted the need to examine the relationship between external knee adduction moments and medial joint contact forces across subjects and activities [17]. With global trends suggesting an increasing prevalence of osteoarthritis and a shift to younger age groups [18], our study aims to contribute to ongoing research efforts to better understand the exact processes that occur within the knee joint and develop effective interventions for improving joint health and function. Given that both positional data and external loads are dynamically defined and measured across various trajectories, the dataset we provide can be effectively employed for the development and validation of dynamic knee models and contribute to the creation of a digital twin of the knee joint.

Given this context, the primary objectives of our study are to develop and validate a novel robotic testbench for studying knee joint biomechanics, with a specific focus on various loading conditions, including varus–valgus and internal–external loading across different flexion angles. By leveraging the capabilities of the robotic testbench, our study aims to investigate the effects of implant positioning in TKA, providing valuable insights that can contribute to improved surgical planning. Ultimately, the development of this advanced testing system can serve as a valuable tool for advancing the understanding of knee joint biomechanics with and without implantation [19].

In recent years, the use of robotic testbenches and cadaver studies has emerged as a promising approach for investigating various aspects of knee joint biomechanics [20,21,22,23]. These systems have been employed to study the motion of total knee replacements during activities such as step-up tasks [20], analyze the effects of polyethylene contact stresses, articular congruity, and knee alignment [21], and measure in vivo tibial forces after TKA using cadaver specimens [24]. Researchers have also examined in vivo contact kinematics and contact forces during dynamic weight-bearing activities [25], determined in vivo kinematics in a multicenter study [26], and presented complete data of total knee replacement loading for level walking and stair climbing measured in vivo with a follow-up of 6–10 months [27]. To build anatomic finite element (FE) models of the lower extremity, static, quasistatic, and dynamic data measured from cadaveric experiments have been considered [28]. Discrete element models based on literature data have also been developed using the OpenSim 4.0 software [29,30]. While these models have significantly advanced our understanding of knee joint biomechanics, their applicability to new measurements from different specimens or patients remains limited due to their fixed geometry and parameterization. The development of computational models, such as the three-dimensional finite element model of the human anterior cruciate ligament by Song et al. [15], has further contributed to the study of knee joint structures. Additionally, Maggioni et al. [31] introduced a bio-inspired robotic testbench designed for repeatable and safe testing of rehabilitation robots, emphasizing the importance of consistent testing environments. Singh et al. [32] highlighted the potential of robotic assistance in TKA procedures, showcasing the precision and versatility of robotic-assisted systems in orthopedic surgeries.

In this study, we augment the biomechanical knowledge base with a unique robotic testbench for comprehensive knee joint analysis. Our testbench, equipped with six degrees of freedom and force–torque control, can execute a wide array of tasks, enabling us to study intricate knee joint interactions under diverse loading conditions. The testbench’s high measurement and control resolution of 2500 Hz ensures robust control and high-quality data, crucial for precise model building and evaluation. Furthermore, our testbench captures all necessary data for model building, including force–torque data on the femur and all positions and orientations. This comprehensive data collection provides a complete picture of knee joint biomechanics, enhancing the accuracy and reliability of potential models. We release the source code for evaluation as well as the collected data for model building at https://github.com/NikonPic/TKA_Testbench (accessed on 1 July 2023).

## 2. Material and Methods

In the following chapter, we delve into our methodology, including the development of a 6 DOF robotic testbench, preparation of cadaveric knee specimens, and execution of a standard TKA procedure. We discuss our custom-designed femoral shields and our approach to biomechanical assessment. We conclude with an overview of our data analysis and outcome measures, providing a succinct summary of our research process.

### 2.1. Robotic Testbench for Knee Joint Biomechanics Assessment

The 6 DOF robotic testbench, as shown in Figure 1, is based on the experimental concept of [33] and extended by [34,35]. The testbench utilizes a robot from the Stäubli RX90B series, which operates at a control frequency of 2500 Hz, making it suitable for dynamic investigations. This robot is further enhanced with a synchronized six-degree-of-freedom (6-DOF) force–torque sensor, ensuring precise control tasks and accurate data acquisition related to knee joint loading. During the tests on the human cadaver knee joint, such as varus–valgus and internal–external loading assessments at varying flexion angles, the robot consistently logs its position and orientation. These recorded measurements form the foundation of the test protocol and are pivotal for the ensuing analysis.

### 2.2. Cadaveric Knee Specimen Preparation and TKA Procedure

After obtaining approval from the local research ethics committee (226/18 S), a fresh-frozen left knee from a 75-year-old male cadaver was prepared for biomechanical testing. First, the Custom 3D-printed mounting plates were attached to the exposed femur and tibia bone surfaces using epoxy resin, ensuring stable geometry for the knee joint, which was then attached to the robotic testbench.

Subsequently, the native knee joint was fixed in the test robot and the testing was performed. Then the second part of the preparation took place, during which a standard TKA procedure was performed on a fresh-frozen human cadaveric knee joint, utilizing a modular implant for the distal femur. The surgical process began with an incision over the front of the knee and opening the joint capsule in terms of a medial arthrotomy to expose the femur, tibia, and patella (Figure 2a). The alignment and resection of the proximal tibia were performed first (Figure 2b), followed by the distal femur (Figure 2c), ensuring proper preparation for the implant components.

Trial components were temporarily inserted to assess the fit, alignment, and stability of the knee joint. Subsequently, the tibial component was implanted. The tibial plateau was prepared to create a flat surface, and the appropriate size of the tibial component was determined using a sizing template. Instead of directly attaching the femoral component to the distal femur, a modular adapter implant was secured to the bone using bone cement. This modular adapter allowed for the attachment of different femoral shields, each with an internal–external rotation of −5° (internal), 0°, and +5° (external), as visualized in Figure 2d. All implants allowed for internal–external rotation during knee flexion–extension motions.

Following the implantation of the adapter (Figure 2d), various femoral shields with different rotational orientations were attached to the adapter (Figure 2e). Prior to initiating the biomechanical tests, the joint capsule and skin incision were closed using sutures. Subsequently, robotic tests were conducted, and the femoral shield was switched between the different rotational configurations to assess the effects of implant variations on knee joint biomechanics. For each switch of implant placement, the knee space was carefully reopened and subsequently resealed. Finally, the cadaveric knee joint was stored at −30 °C in an airtight environment to ensure their preservation and prevent any degradation.

### 2.3. Femur Shield Design

To investigate the impact of different rotations on cadaveric knee joints, customized femoral implant shields were designed, 3D-printed, and mounted on an adapter. The basis for these implants was derived from segmented CT scans of the knee joint. Segmentation of the tibia, fibula, patella, and femur was performed using 3D Slicer 5.0.3 software [36], as illustrated in Figure 3a. The segmented tibia and femur data provided the foundation for TKA planning, determining the appropriate implant sizes. As a reference [37], Triathlon TKA implants were modified to match the obtained knee dimensions. The resulting meshes were then 3D-printed using the [38] v2 printer.

For the femoral shield design, the obtained femoral implant was rotated to achieve internal/external rotations of −5° (internal), 0° (neutral), and +5° (external). This step was executed in [39] 3.0 software, as demonstrated in Figure 3b. This custom approach allowed for a precise evaluation of the effects of various implant rotations on knee joint biomechanics.

### 2.4. Assessing Knee Joint Biomechanics: Experimental Approach

A comprehensive biomechanical assessment of the cadaveric knee joint was conducted using the robotic testbench system. The evaluation involved varus–valgus and internal–external loading tests at multiple flexion angles, specifically at 0°, 30°, 60°, and 90°, for both the native knee joint and TKA conditions with varying implantation rotations. The rate of rotation to achieve the 5 Nm torque was maintained consistently at 0.5°/s. To ensure uniformity and consistency in the testing procedure, tests were performed in a specific order of knee flexion for each varus–valgus and internal–external rotation. Furthermore, to minimize any potential order effect, the sequence of tests was randomized.

Force–torque data, position, and quaternion information were meticulously collected throughout the experiment. This data collection facilitated an in-depth analysis of the knee joint’s biomechanical response under diverse loading conditions and allowed for robust comparisons between the native knee joint and the various TKA conditions. Tests were executed along the defined axis, up to a maximum load of 5 Nm. Each specific loading and flexion condition was repeated three times independently, with results captured at a 2500 Hz sampling rate. From these three independent tests, mean values and standard deviations were computed to provide a comprehensive overview of the biomechanical behavior of the knee joint under the tested conditions.

### 2.5. Data Analysis and Outcome Measures

The data collected during the experimental procedures, including force–torque data, position, and quaternion information, were processed and analyzed to evaluate the biomechanical properties of the native knee joint and TKA conditions. The following outcome measures were assessed:Robotic testbench reproducibility: The precision and reproducibility of the robotic testbench were verified through repeated tests under identical conditions. The recorded force–torque data and position–quaternion information provided high-accuracy measurements of the biomechanical parameters.Knee joint stability: The stability of the knee joint under various loading conditions, including varus–valgus and internal–external loads, was evaluated by analyzing the force–torque data and corresponding joint displacement.Comparison between native and TKA conditions: The biomechanical properties of the native knee joint and the knee joint with TKA were compared to determine the effects of the arthroplasty procedure on joint function and stability.Effects of varying femoral component rotations: The impact of different femoral component rotations in TKA on the knee joint’s biomechanical properties was examined by comparing the joint stability and kinematic properties under femoral component rotations of −5° (internal), 0° (neutral), and +5° (external).

Statistical analysis was conducted to identify significant disparities between the native knee and TKA conditions, and among varying femoral component rotations in TKA. The outcomes of this investigation enrich our understanding of knee joint biomechanics and the implications of femoral component variations for joint stability and function. This comprehension may guide surgical strategies and prosthesis design, potentially enhancing post-TKA patient outcomes.

## 3. Results

### 3.1. The Testbench

The real test performed and the measurement results are shown in Figure 4. The clamped specimen knee joint was tested with the robot and provided motion and force–torque data first in native condition and later in the specific TKA configuration. Key components of the experimental setup include femur fixation, force–torque sensor, and trackers for both the femur and tibia (labeled in Figure 4a).

The Stäubli RX90B robot, attached to the native knee specimen, facilitates a comprehensive evaluation of knee joint biomechanics under various loading scenarios (Figure 4a). Figure 4b–d demonstrate the impact of different forces and torques on knee joint function and stability. Figure 4b highlights the testing system’s precision, while Figure 4c,d present the measured forces and corresponding torques. This experimental approach has the potential to enhance our understanding of knee joint biomechanics and inform improved treatment strategies for knee-related pathologies.

### 3.2. Analysis of Biomechanical Variations in Knee Joint with Different TKA Configurations

Figure 5 provides a visual representation of the maximum deviation angles and their corresponding standard deviations for both internal–external and varus–valgus loading at flexion angles of 10°, 20°, 30°, 60°, and 90°. The robotic testbench performed each test three times, applying 5 Nm of torque. Different colors and symbols are used in the figure to distinguish the native knee and TKA configurations. A legend is provided for clarity.

For varus (Figure 5a) and valgus (Figure 5b) loading, the native knee demonstrates superior stability, exhibiting the lowest deviations (<2°) from the neutral position, which corresponds to zero loading along the respective axis. In contrast, all TKA configurations exhibit diminished stability, with the −5° TKA displaying the lowest stability for varus loading at 90° flexion (>6°). The robotic testbench achieves high precision, with standard deviations below 0.7° (maximum standard deviation for 90° flexion and valgus loading for −5° TKA: 0.65°).

Analyzing the internal (Figure 5c) and external loading (Figure 5d) results reveals greater variability between the native knee and TKA variants. Unlike varus loading, the −5° TKA exhibits the highest stability for internal and external loading. The +5° TKA most closely approximates the native knee’s stability, while the −5° TKA demonstrates the greatest deviation. The testbench’s precision for internal–external loading tasks is superior, with standard deviations below 0.1°.

### 3.3. Stability Evaluation

The Total Deviation Angle (TDA), determined as the total sum of angles between extreme positions under specific loading conditions, serves as a measure of joint instability. Higher TDA values correspond to reduced joint stiffness under the corresponding loading scenario. Figure 6 presents the differential impacts of various TKA modifications, alongside the native knee, on joint stability across diverse loading conditions. The Kiviat diagram describes TDA in the context of internal/external (Figure 6b) and varus/valgus (Figure 6a) loading across a spectrum of flexion angles under a 5 Nm load, contrasting the native knee with three TKA orientations (neutral 0°, −5° internal rotation, and +5° external rotation of the femoral component).

The native knee demonstrates the greatest stability with minimal deviations, whereas the TKA variations exhibit increased deviations in both loading conditions. The −5° TKA variation, in particular, exhibits the lowest stability in varus–valgus loading at 90° flexion, while for internal–external loading, it displays the highest stability. Conversely, the +5° TKA closely approximates the native knee performance, with the exception of the −5° TKA exhibiting the highest deviation. Error bars in the figure represent the standard deviation, highlighting the precision of the robotic testbench. This analysis underscores the importance of TKA variation strategies in maintaining knee joint biomechanics and stability under varied loading conditions.

Figure 6 presents Kiviat diagrams of the Total Deviation Angle (TDA) as a measure for instability in internal/external and varus/valgus loading for the native knee and various Total Knee Arthroplasty (TKA) variations (neutral 0°, −5° internal, and +5° external) across different flexion angles under 5 Nm loading conditions. The diagrams are divided into two parts: (a) varus/valgus loading and (b) internal/external loading. These diagrams effectively illustrate the sum of the deviations for varus/valgus and internal/external loading, thereby providing a comprehensive view of the impact of different TKA variation strategies on knee joint biomechanics. The radial nature of the Kiviat diagrams allows for an easy comparison of the TDA across different loading conditions and TKA variations, highlighting the biomechanical differences and potential instabilities introduced by these variations. Notably, the value for the −5° internal TKA at 20° flexion was determined through linear interpolation between the values at 10° and 30° flexion.

### 3.4. Comparison of TKA Variations to the Native Knee Joint

In this section, we perform a quantitative analysis of the differences between the native knee and various TKA configurations under varus, valgus, internal, and external loading conditions. We use the Mean Squared Error (MSE) to evaluate the deviation angles of the TKA variants across all flexion angles and loading conditions, using the native knee joint as a reference. These deviations are critical because they reveal the extent of the discrepancy between the native knee joint and its prosthetic replacement. Understanding these differences can have profound implications for clinical outcomes, including patient satisfaction, joint stability, and implant longevity.

Table 1 presents the MSE values that quantify the deviation between the native knee and the different TKA configurations under varus and valgus loading conditions. The TKA with a neutral (0°) femur shield orientation exhibited the lowest mean deviation for both varus (5.51 ± 4.98) and valgus (0.88 ± 0.76) loading, implying superior biomechanical similarity to the native knee under these conditions. Conversely, the TKA with a −5° internal rotation of the femur shield presented the highest mean deviation for both varus (11.36 ± 14.04) and valgus (2.05 ± 1.55) loading, which could potentially be associated with less desirable clinical outcomes. The overall minimal mean deviation was observed for the TKA with a neutral (0°) femur shield rotation, demonstrating a value of 3.20 ± 2.57.

Table 2 presents the MSE values, quantifying deviations between the native knee and distinct TKA configurations under internal and external loading conditions. The TKA with a +5° external rotation of the femur shield exhibited the minimal mean deviation for both internal (7.78 ± 9.70) and external (5.96 ± 4.23) loading, suggesting that this TKA variation may promote improved joint stability under these conditions in comparison to other configurations. In contrast, the TKA with a −5° internal rotation of the femur shield showed the maximal mean deviation, with values of 81.97 ± 105.40 for internal and 41.22 ± 58.30 for external loading, potentially implicating unfavorable clinical consequences. The overall lowest mean deviation for combined internal and external loading was observed for the TKA with a +5° external rotation of the femur shield, displaying a value of 6.87 ± 4.83.

## 4. Discussion

### 4.1. TKA Variations: Biomechanical Compatibility

This study focused on the biomechanical compatibility of diverse TKA positions under different loading conditions, with focus on the stability of the joint in the various positions. The objective was to discern how divergences between the native knee and TKA variations transpire under varus and valgus loading conditions, in addition to internal and external loading scenarios.

Our data revealed that the TKA modification with the femoral component at neutral 0° manifested the minimum mean deviation under both varus and valgus loading, implying superior biomechanical compatibility for these conditions. Conversely, the −5° variant demonstrated the maximum mean deviation, indicating potentially less-than-ideal clinical outcomes. For internal and external loading conditions, the TKA modification with +5° rotation displayed the lowest mean deviation, suggesting enhanced joint stability compared to the other configurations. The −5° variant displayed the highest mean deviation, suggesting less desirable clinical outcomes.

### 4.2. TKA Modification Strategies: Comparison with Prior Research

Our analysis demonstrated that the TKA variation with the femoral component at neutral 0° displayed the lowest mean deviation under varus and valgus loading, while the +5° variation yielded the lowest mean deviation under internal and external loading. This suggests that specific TKA modifications may provide optimal biomechanical compatibility and joint stability under different loading conditions. The results of our analysis demonstrated that the TKA variation with the femoral component at neutral 0° displayed the lowest mean deviation under varus and valgus loading, while the +5° variation yielded the lowest mean deviation under internal and external loading. Based on these findings, an estimated “native zone” for optimal biomechanical compatibility and joint stability could be estimated between 0° and +5°.

Earlier research by [40,41,42] reported promising outcomes using patient-specific alignment in TKA, with the goal of restoring native joint alignment and kinematics. Our findings endorse these observations, as the TKA modification with the femoral component at neutral 0°, which is closer to patient-specific alignment, exhibited superior performance under varus and valgus loading. Rivière et. al. [43] undertook a systematic review of TKA modification strategies and inferred that a personalized approach might be most effective. Our findings support this proposal, as the TKA variation with +5° rotation demonstrated superior joint stability under internal and external loading, emphasizing the possible advantages of a patient-specific approach based on individual loading conditions. While contrasting our findings with [44], which reported no significant difference in 2-year functional outcomes between kinematic and mechanical alignment, it is crucial to highlight that our study concentrates on biomechanical compatibility and joint stability across diverse loading conditions. While our results suggest certain TKA modifications may deliver superior biomechanical outcomes, factors such as patient satisfaction and implant durability also play a vital role when gauging the overall clinical success.

Collectively, our findings align with past research advocating the application of individualized TKA modification strategies to optimize biomechanical compatibility and joint stability. Future research is warranted to evaluate the long-term clinical outcomes related to these modification strategies in TKA.

### 4.3. Biomechanical Implications and Potential of Robotic Testbench Applications

This study furnishes crucial insights into the biomechanics of various TKA modifications and their effects on the native knee joint. Understanding the discrepancies between the native knee and TKA variations under diverse loading scenarios is integral to refining surgical procedures, ensuring patient satisfaction, joint stability, and enhancing implant durability.

The robotic testbench utilized in this study demonstrates potential for broad application in biomechanical research. Its capacity to accurately reproduce and manage diverse loading conditions and simulate complex knee joint kinematics renders it a valuable resource for assessing different TKA variations, surgical procedures, and implant designs. The testbench can be deployed to investigate the influence of various factors on knee joint biomechanics, including age, gender, and pre-existing conditions, facilitating a more in-depth comprehension of personalized surgical planning and bespoke implant design.

Importantly, the robotic testbench’s ability to emulate intricate knee joint interactions under controlled conditions provides a foundation for extrapolating its results to in vivo arthroplasty procedures. By simulating real-life loading scenarios and knee joint kinematics, the testbench offers insights into how different TKA variations and implant designs might perform in a living body. This controlled environment allows for a systematic evaluation of biomechanical effects, which can then be applied to predict in vivo performance, potentially leading to improved surgical outcomes and patient-specific implant designs.

Furthermore, the robotic testbench could serve in pre-clinical evaluations of novel implant designs, allowing researchers to measure their biomechanical performance under various loading conditions prior to initiating clinical trials. This approach may expedite the development of more effective and durable implants, reducing the need for revision surgeries and improving patient outcomes.

### 4.4. Limitations

Several constraints are inherent in this study, which ought to be acknowledged when interpreting the results. A prominent limitation is that the testbench and TKA variations were assessed using a single knee specimen, which restricts the extrapolation of findings to a wider population. The focus of the study was primarily on biomechanical compatibility and joint stability under diverse loading conditions in direct comparison, omitting consideration of factors like patient satisfaction, implant durability, and the influence on soft tissues. Furthermore, this investigation did not assess the long-term clinical outcomes linked with different TKA modifications. In addition, this study did not incorporate traditional alignment strategies and was solely limited to variations in femoral component rotations. This limitation may omit important factors related to alignment strategies that could influence the results and their clinical implications.

Though the robotic testbench facilitated exact control and replication of loading scenarios, it may not wholly emulate the intricate in vivo environment of the knee joint. Future investigations should strive to overcome these limitations, exploring the impacts of TKA modifications on a larger and more diverse population. Such studies would provide a more exhaustive understanding of the selection of TKA modifications and their repercussions on clinical outcomes.

This study’s significant limitation is the reliance on a single knee specimen to evaluate the testbench and TKA variations, which curtails the broad applicability of our findings. The inherent variability in knee biomechanics across individuals, influenced by factors such as age, gender, injuries, and anatomical differences, suggests that a single specimen might not adequately represent the wider population. To achieve a more comprehensive understanding in future research, it would be beneficial to utilize multiple knee specimens from a variety of donors, factor in specimens with pre-existing conditions or injuries, and collaborate with several institutions to bolster specimen diversity and sample size. By addressing these considerations, we can enrich our grasp of knee biomechanics, leading to more informed clinical decisions and improved patient outcomes.

## 5. Conclusions

This study highlights the pivotal influence of TKA positioning on biomechanical compatibility and joint stability. Specifically, TKA with ±0 degrees of rotation excels under varus and valgus loading, while +5 degrees of rotation are optimal for internal and external loading. These results underscore the importance of tailored TKA selection to enhance clinical outcomes. The robotic testbench emerges as a valuable tool in biomechanical research, offering insights into diverse TKA variations and paving the way for improved surgical approaches and patient outcomes post-total knee arthroplasty.

## Figures and Tables

**Figure 1 sensors-23-07459-f001:**
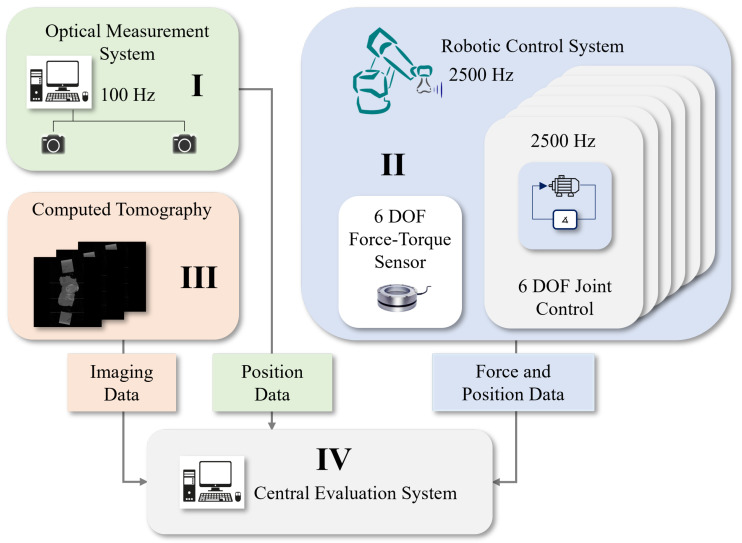
Testbench architecture consisting of an optical measuring system (**I**), the 6 DOF Robot Stäubli RX90B (**II**) with an additional 6 DOF force–torque sensor for external load acquisition of the respective joint, CT imaging (**III**), and the central evaluation system (**IV**).

**Figure 2 sensors-23-07459-f002:**
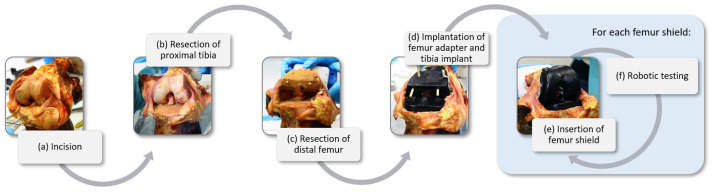
Sequential steps of the modified total knee arthroplasty (TKA) procedure on a cadaveric knee joint: (**a**) opening of the joint capsule to expose the femur, tibia, and patella; (**b**) resection of the proximal tibia; (**c**) resection of the distal femur; (**d**) implantation of the tibial component and femoral adapter; and (**e**) switching femoral shields with different internal–external rotations (−5°, 0°, and +5°) during biomechanical testing (**f**).

**Figure 3 sensors-23-07459-f003:**
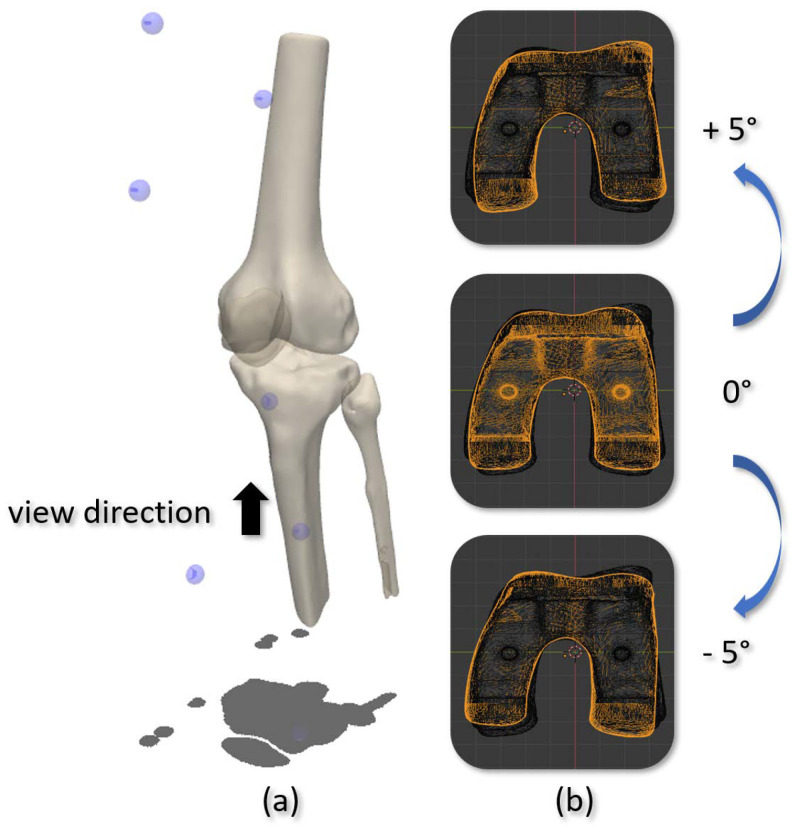
(**a**) 3D Slicer segmentation of the cadaveric knee joint, demonstrating the tibia, fibula, patella, and femur, along with the femoral (upper blue spheres) and tibial (lower blue spheres) trackers. (**b**) Tailored femoral component designs modeled on Triathlon CR ([37] GmbH, Duisburg, Germany) implants, resized and rotated to −5° (internal), 0° (neutral), and +5° (external). The femoral components are displayed on the transverse (axial) plane, viewed distally, as indicated by the direction of the black arrow in (**a**).

**Figure 4 sensors-23-07459-f004:**
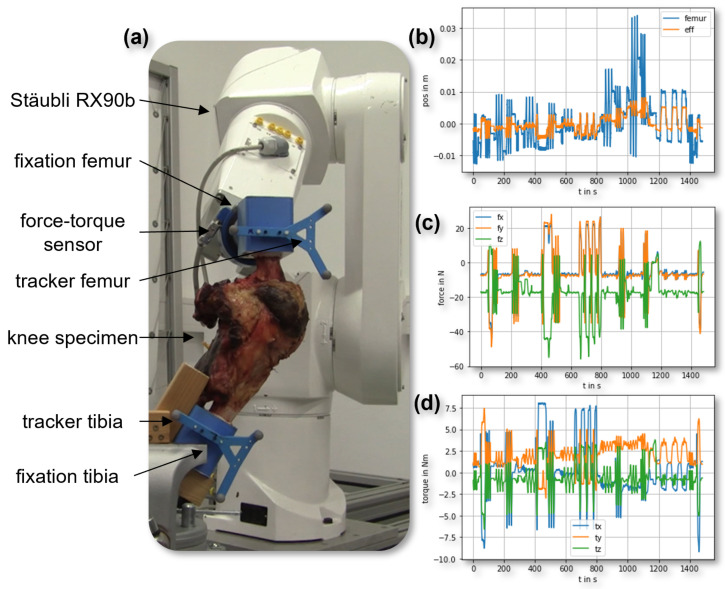
Visualization of the complete test setup (**a**), the synchronization and movement of the femur tracker (femur, blue) and the end-effector of the robot (eff, orange) (**b**), as well as the acquired force (**c**) and torque (**d**) data during the test of the native knee.

**Figure 5 sensors-23-07459-f005:**
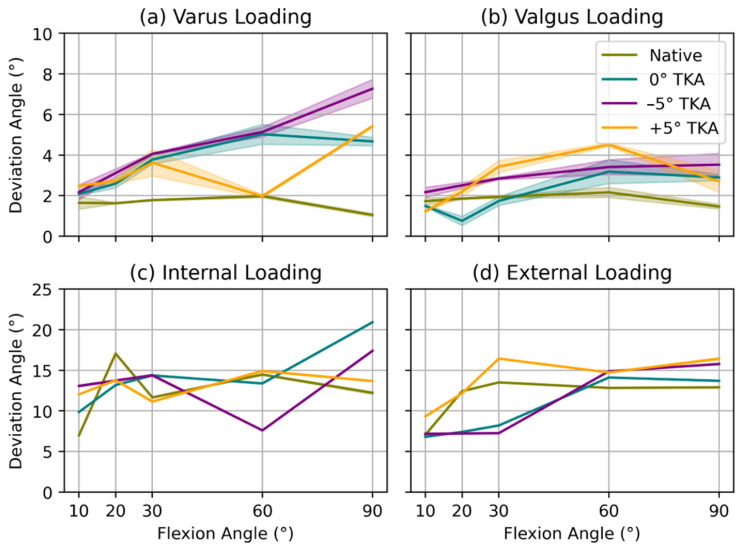
Comparison of the maximum deviation angles and their corresponding standard deviations for (**a**) varus, (**b**) valgus, (**c**) internal, and (**d**) external loading at 5 Nm and three repetitions of the test. The measurements were taken at distinct flexion angles of 10°, 20°, 30°, 60°, and 90° for both the native knee and total knee arthroplasty (TKA) configurations with 0°, −5°, and +5° implant rotations.

**Figure 6 sensors-23-07459-f006:**
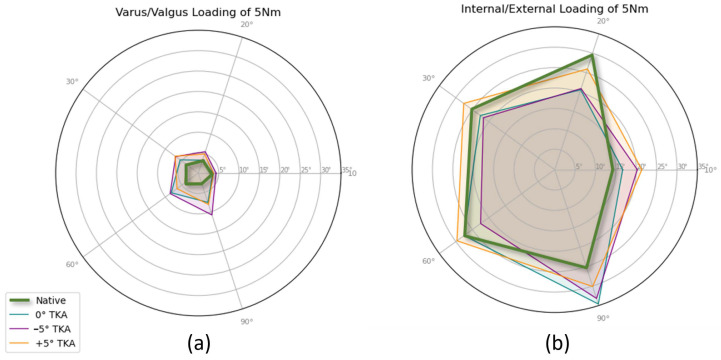
Kiviat Diagram of the Total Deviation Angle (TDA) as a measure for instability in internal/external and varus/valgus loading for native knee and various TKA variations (neutral 0°, −5° internal, and +5° external) across different flexion angles under 5 Nm loading conditions. (**a**) Varus/valgus loading and (**b**) internal/external loading. The presented results display the sum of the deviations for varus/valgus and internal/external loading, illustrating the impact of different TKA variation strategies on knee joint biomechanics.

**Table 1 sensors-23-07459-t001:** Difference of angular deviations in degrees between the native knee and different TKA for varus and valgus loading evaluated by mean squared error (MSE) and the respective standard deviations. Values in bold indicate the smallest deviation from the native knee.

	Angular Deviations to Native in °		
TKA	Varus	Valgus	Mean
+5°	**4.87** ± 7.19	1.93 ± 1.97	3.40 ± 3.67
±0°	5.51 ± 4.98	**0.88** ± 0.76	**3.20** ± 2.57
−5°	11.36 ± 14.04	2.05 ± 1.55	6.71 ± 7.14

**Table 2 sensors-23-07459-t002:** Difference of angular deviations in degrees between the native knee and different TKA for internal and external loading evaluated by mean squared error (MSE) and the respective standard deviations. Values in bold indicate the smallest deviation from the native knee.

	Angular Deviations to Native in °		
TKA	Internal	External	Mean
+5°	**7.78** ± 9.70	**5.96** ± 4.23	**6.87** ± 4.83
±0°	21.50 ± 27.44	11.14 ± 12.73	16.32 ± 18.03
−5°	81.97 ± 105.40	41.22 ± 58.30	61.60 ± 74.86

## Data Availability

The data presented in this study are openly available and can be found in the GitHub repository at https://github.com/NikonPic/TKA_Testbench. (accessed on 1 July 2023).

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
