# Peer review of "Validation of a Robotic Testbench for Evaluating Biomechanical Effects of Implant Rotation in Total Knee Arthroplasty on a Cadaveric Specimen"

_sensors, 2023, doi:10.3390/s23177459_

Round 1

Reviewer 1 Report

This paper presents a robotic testbech to sudy the biomechanical compatibility for total knee arthroplasty under various loading configurations. The paper is well written and presented. Som minor issues are the follwing:

1. How this robotic testbench can help to extrapolate its results to in-vivo arthroplasty procedures?

2. Please avoid subjective sentences such as: 

Line 127: Subsequently,....in standard procedure. Which standar is followed?

Line 138: ...was stored appropiately.

3.- It might be difficult to reproduce this testbench as presented. Is it possible to render more generic components to give advice other research gruoups on how to build a copy?

Author Response

Dear reviewer,

We sincerely appreciate your insightful comments and constructive feedback. We have made every effort to address each of your concerns comprehensively.

  1. How this robotic testbench can help to extrapolate its results to in-vivo arthroplasty procedures?

    Thank you for that comment. This point has not been dealt with in sufficient detail. We have therefore added the following paragraph to the discussion:

    Importantly, the robotic testbench's ability to emulate intricate knee joint interactions under controlled conditions provides a foundation for extrapolating its results to in-vivo arthroplasty procedures. By simulating real-life loading scenarios and knee joint kinematics, the testbench offers insights into how different TKA variations and implant designs might perform in a living body. This controlled environment allows for a systematic evaluation of biomechanical effects, which can then be applied to predict in-vivo performance, potentially leading to improved surgical outcomes and patient-specific implant designs.

  2. Please avoid subjective sentences such as: 

    Line 127: Subsequently,....in standard procedure. Which standard is followed?

    Thank you for the detailed analysis on misplaced subjective sentences. For the remark on Line 127, we have added the following information to the text:

    Subsequently, the tibial component was implanted. The tibial plateau was prepared to create a flat surface, and the appropriate size of the tibial component was determined using a sizing template.

    Line 138: ...was stored appropiately.
    Thank you for pointing out the need for clarity regarding storage methods. To address your concern, the knee joint specimens were stored at -30°C in an airtight environment to ensure their preservation and prevent any degradation. This method aligns with best practices for preserving biological specimens. We will include this specific detail in the revised manuscript to provide a clear understanding of our storage protocol.

  3. It might be difficult to reproduce this testbench as presented. Is it possible to render more generic components to give advice other research gruoups on how to build a copy?

    Thank you for highlighting the importance of reproducibility in our work. We acknowledge that the specific components and configurations of our testbench might be challenging for some research groups to replicate precisely. Therefore, we have decided to further provide all necessary information on the testbench archticture and STL files in the github repository to enable a simplified transition in replication the testbench using the provided framework.

    https://github.com/NikonPic/TKA_Testbench

Once again, thank you for your invaluable feedback. We believe that your suggestions have significantly enhanced the quality of our manuscript.

Best regards,
Nikolas Wilhelm

Reviewer 2 Report

1. The abstract could include more results in the manuscript, and it would be more convenient for the reader to get more information.

2. The authors have said that "In recent years, the use of robotic test benches and cadaver studies has emerged as 

a promising approach for investigating various aspects of knee joint biomechanics"(lines 68-69).  However, the authors did not say much about "Robotic Testbench", and it is better to add the advance in robotic test bench, and some examples are as follows.

1)Maggioni, S., Stucki, S., Lünenburger, L., Riener, R., & Melendez-Calderon, A. (2016, June). A bio-inspired robotic test bench for repeatable and safe testing of rehabilitation robots. In 2016 6th IEEE International Conference on Biomedical Robotics and Biomechatronics (BioRob) (pp. 894-899). IEEE.

2)Singh, V., Teo, G. M., & Long, W. J. (2021). Versatility and accuracy of a novel image-free robotic-assisted system for total knee arthroplasty. Archives of Orthopaedic and Trauma Surgery, 141(12), 2077-2086.

3. There are some tests on the human cadaver knee joint, and the detail of the human cadaver could be shown in the manuscript, such as age, gender, and so on, because it could affect the results.

4. In the title of Figure 3, it is said "The femoral components are presented on the axial plane, viewed from the distal end." It is better to mark the view direction by the arrow in the picture.

5.  In Figure 6, picture (a) is too small, and it is difficult to get details in this picture(a).

6. In Table 1, The table is well-formatted, but some data have no units. 

7. In the Discussion part, the manuscript said that "A prominent limitation is that the testbench and TKA variations were assessed using a single knee specimen"(line 346). It is better for the author to explain to the reader how it can be improved in the future. Because this is a common phenomenon in this field, everyone is more concerned.

Author Response

Dear Reviewer 2,

We sincerely appreciate your thorough review and constructive feedback on our manuscript. Your insights have been invaluable, and we have made concerted efforts to address each of your points:

1. The abstract could include more results in the manuscript, and it would be more convenient for the reader to get more information.

Thank you for the remark. We have extended the abstract accordingly, by providing more information and more results, while keeping the overall abstract size small:

The native knee showed enhanced stability in varus-valgus loading, with the +5° external rotation TKA displaying the smallest deviation, indicating biomechanical compatibility. The robotic testbench consistently demonstrated high precision across all loading conditions.

2. The authors have said that "In recent years, the use of robotic test benches and cadaver studies has emerged as a promising approach for investigating various aspects of knee joint biomechanics"(lines 68-69).  However, the authors did not say much about "Robotic Testbench", and it is better to add the advance in robotic test bench, and some examples are as follows.

1)Maggioni, S., Stucki, S., Lünenburger, L., Riener, R., & Melendez-Calderon, A. (2016, June). A bio-inspired robotic test bench for repeatable and safe testing of rehabilitation robots. In 2016 6th IEEE International Conference on Biomedical Robotics and Biomechatronics (BioRob) (pp. 894-899). IEEE.

2)Singh, V., Teo, G. M., & Long, W. J. (2021). Versatility and accuracy of a novel image-free robotic-assisted system for total knee arthroplasty. Archives of Orthopaedic and Trauma Surgery, 141(12), 2077-2086.

Thank you very much for the provided literature and the remark to highlight the advances in robotic testbenches. We have extended the paragraph about knee joint biomechanics accordingly:
... contributed to the study of knee joint structures. Additionally, Maggioni et al. [1] introduced a bio-inspired robotic test bench designed for repeatable and safe testing of rehabilitation robots, emphasizing the importance of consistent testing environments. Singh et al. [2] highlighted the potential of robotic assistance in TKA procedures, showcasing the precision and versatility of robotic-assisted systems in orthopedic surgeries.

3. There are some tests on the human cadaver knee joint, and the detail of the human cadaver could be shown in the manuscript, such as age, gender, and so on, because it could affect the results.

Thank you for this advice. The human cadaver was a male of age 75 and the prargraph containing the relevant information was changed as follows:

After obtaining approval from the local research ethics committee (226/18 S), a fresh-frozen left knee from a 75-year-old male cadaver was prepared for biomechanical testing.

4. In the title of Figure 3, it is said "The femoral components are presented on the axial plane, viewed from the distal end." It is better to mark the view direction by the arrow in the picture.

Thank you very much for this clever idea. We have added an arrow in Figure3 (a) and explained in the caption accordingly:
The femoral components are displayed on the axial plane, viewed distally, as indicated by the direction of the black arrow in (a).

5.  In Figure 6, picture (a) is too small, and it is difficult to get details in this picture(a).

Thank you for paying attention to this detail. The Figure is indeed too small and we have therefore enlargened the figure accordingly.

6. In Table 1, The table is well-formatted, but some data have no units. 

That is indeed true, the units are not clearly labeled in this table, thank you for paying attention to tis detail. We have provided the units [angular degrees] to the caption and the table itself:

Caption: Difference of angular deviations in degrees between the native knee and different TKA for varus and valgus loading evaluated by mean squared error (MSE) and the respective standard deviations.
Table: Angular Deviations to Native in °

7. In the Discussion part, the manuscript said that "A prominent limitation is that the testbench and TKA variations were assessed using a single knee specimen"(line 346). It is better for the author to explain to the reader how it can be improved in the future. Because this is a common phenomenon in this field, everyone is more concerned.

Thank you for your insightful feedback and for highlighting the importance of addressing the limitation of using a single knee specimen in our study. We have tried to further address this in the limitations section and added the following paragraph to this section:

This study's significant limitation is the reliance on a single knee specimen to evaluate the testbench and TKA variations, which curtails the broad applicability of our findings. The inherent variability in knee biomechanics across individuals, influenced by factors such as age, gender, injuries, and anatomical differences, suggests that a single specimen might not adequately represent the wider population. To achieve a more comprehensive understanding in future research, it would be beneficial to utilize multiple knee specimens from a variety of donors, factor in specimens with pre-existing conditions or injuries, and collaborate with several institutions to bolster specimen diversity and sample size. By addressing these considerations, we can enrich our grasp of knee biomechanics, leading to more informed clinical decisions and improved patient outcomes.

Once again, thank you for your thoughtful feedback. We believe your suggestions have significantly enhanced the manuscript's quality and rigor.

Best regards,

Nikolas Wilhelm

Reviewer 3 Report

The study is interesting and presents a valuable line of research. It provides valuable insights into the biomechanical compatibility of various total knee arthroplasty configurations, shedding light on their influence on knee joint stability.

I have some coments:

Material and Methods

No specific details regarding patient consent. Your manuscript does not contain a complete IRB statement regarding ethics board approval. Original articles need to contain a statement about the Helsinki Declaration of 1975, as in the example given here: “This study was approved by the human subjects ethics board of XXXXX and was conducted in accordance with the Helsinki Declaration of 1975, as revised in 2013.

Conclusions

While the conclusion presents important findings, it appears to be excessively lengthy. To enhance its clarity and impact, it would be beneficial to condense the key points and focus on the most significant outcomes.

Author Response

Dear Reviewer 3,

We sincerely appreciate your thorough review and the constructive feedback provided. Your insights have been instrumental in refining our manuscript, and we have addressed each of your comments as follows:

No specific details regarding patient consent. Your manuscript does not contain a complete IRB statement regarding ethics board approval. Original articles need to contain a statement about the Helsinki Declaration of 1975, as in the example given here: “This study was approved by the human subjects ethics board of XXXXX and was conducted in accordance with the Helsinki Declaration of 1975, as revised in 2013.

Thank you very much for paying attention to this critical point. Indeed the ethics approval was not sufficiently mentioned in the manuscript. We have tried to correct this by providing the following declaration in the manuscript both in text as well as at the end:

In text: After obtaining approval from the local research ethics committee (226/18 S), a fresh-frozen left knee from a 75-year-old male cadaver was prepared for biomechanical testing.

Institutional Review Board Statement: The study was conducted in accordance with the Declaration of Helsinki, and approved by the Institutional Ethics Committee of Klinikum Rechts der Isar (226/18 S, 29.05.2018).

Conclusions

While the conclusion presents important findings, it appears to be excessively lengthy. To enhance its clarity and impact, it would be beneficial to condense the key points and focus on the most significant outcomes.

Regarding your remark on the conclusion section, we understand your concerns about its length and clarity. We have taken your feedback into account and have revised the conclusion to be more concise, focusing on the most significant outcomes of our research. We believe this revision enhances the clarity and impact of our findings, making it easier for readers to grasp the key takeaways:

This study highlights the pivotal influence of TKA positioning on biomechanical compatibility and joint stability. Specifically, TKA with ±0 degrees rotation excels under varus and valgus loading, while +5 degrees rotation is optimal for internal and external loading. These results underscore the importance of tailored TKA selection to enhance clinical outcomes. The robotic testbench emerges as a valuable tool in biomechanical research, offering insights into diverse TKA variations and paving the way for improved surgical approaches and patient outcomes post-total knee arthroplasty.

Once again, thank you for your invaluable feedback. We believe your suggestions have significantly enhanced the manuscript's quality and rigor.

Best regards,

Nikolas Wilhelm

Reviewer 4 Report

This manuscript focused on the biomechanical effects of diverse TKA positions under different loading conditions on a cadaveric specimen. The results indicated that a TKA variation with 0 degrees rotation offers superior performance for varus and valgus loading, whereas a +5 degrees rotation variation provides enhanced stability for internal and external loading. I think this work helps to gain a deeper understanding of personalized surgical planning and customized implant design. Here are some suggestions that may help with the publication.

1, Figure 1. Is the sensor measuring external force or joint force? If it is the latter, please briefly explain the measurement principle.

2, I think some expressions are not very standardized, such as the content on lines 104 and 105.

3. line170, “The precision and reproducibility of the robotic testbench were verified through repeated tests under identical conditions”. Which part of the article is the result that supports this conclusion located?

4, Finally, in Figure 4b, what does the "eff" line represent?

Author Response

Dear Reviewer,

Thank you for your insightful comments and constructive feedback on our manuscript. We are pleased to hear that you recognize the value of our work in advancing the understanding of personalized surgical planning and customized implant design. We have carefully considered each of your suggestions and would like to address them as follows:

  1. Regarding Figure 1: The sensor measures the load on the joint via a force-torque sensor, which measures external loads. We have clarified this ambiguity by revising the caption as follows:

    "Test bench architecture consisting of an optical measuring system (I), the 6 DOF Robot Stäubli RX90B (II) with an additional 6 DOF force-torque sensor for external load acquisition of the respective joint, CT imaging (III), and the central evaluation system (IV)."

  2. Expressions on lines 104 and 105: We acknowledge your observation about the non-standardized expressions, thank you for pointing this out. We have reviewed these lines and made the necessary corrections to ensure clarity and standardization:

    "The testbench utilizes a robot from the Stäubli RX90B series, which operates at a control frequency of 2500 Hz, making it suitable for dynamic investigations. This robot is further enhanced with a synchronized six-degree-of-freedom (6-DOF) force-torque sensor, ensuring precise control tasks and accurate data acquisition related to knee joint loading."

  3. Regarding the statement on line 170: We apologize for any ambiguity. Rereading the manuscript it is not clear, where the observation for "The precision and reproducibility of the robotic testbench" is supported.
    The results supporting the precision and reproducibility of the robotic testbench can be found in Section 3.2 (Analysis of Biomechanical Variations in Knee Joint with Different TKA Configurations), where we detail the repeated tests and their consistent outcomes, specifically in Figure 5, where low standard deviations troughout all performed measurements can be observed. We have further emphaiszed this in the caption of Figure 5 and extended the text corresponding to Figure 5:
    "The robotic testbench achieves high precision, with standard deviations below 0.7° (maximum standard deviation for 90° flexion and valgus loading for -5° TKA: 0.65°)."

  4. Concerning Figure 4b: The "eff" represents the "end-effector" of the robotic system. As both internal tracking by the robotic joint sensors as well as external tracking of the tracker of the femur are performed, both are required for synchronisation and precise angle measurement. We realize that this was not adequately explained in the original figure caption, and we have now added a detailed description to address this:
    "Visualization of the complete test setup in (a), the synchronization and movement of the femur tracker (femur, blue) and the end-effector of the robot (eff, orange) in (b) as well as the acquired force (c) and torque (d) data during the test of the native knee."

We are grateful for your thorough review and the opportunity to improve our manuscript. We believe that these revisions enhance the clarity and quality of our work. We look forward to any further feedback or recommendations you may have.

Best regards,

Nikolas Wilhelm

Reviewer 5 Report

Summary:

A cadaveric knee was tested without and with three different total knee arthroplasty (TKA) implants to measure the joint stability of the knee at different flexion angles. Intern-external and varus-valgus loads were applied up to 5 Nm with a robotic testbench while the femur and tibia kinematics were tracked.

Comments:

The authors have provided a good assessment of the TKA implants for evaluating knee joint biomechanics. The details, figures, and tables provide good explanations to the reader of the experimental set-up and test-results. Overall, this is a good study to evaluate the effectiveness of different TKA implants in terms of stability and reliability. As the authors mention in the Discussion section, additional cadaveric data are necessary to determine the effectiveness of these implants. Although rotational forces were applied, it would be advantageous to also assess linear forces to determine stress distribution within the knee joint.

Additional Comments:

One point which the authors provide is the comparison between the knee joint mechanics when the knee was tested before and after TKA. In lines 112-126 this is mentioned, but it seems to be buried – meaning the emphasis is on the TKA testing. It is suggested that the authors re-emphasize the native and TKA conditions so the reader/audience can recall these conditions for the comparisons.

When rotating the knee in flexion, from anatomical 0 degrees, the knee joint also pivots slightly about the longitudinal axis (screw-home mechanism) – this can be seen in Figure 6 at 20 deg of knee flexion. Do these implants also for such deviations so that the tibia can internally and externally rotate during the knee flexion-extension motions?

Lines 20-24: the proximal tibiofibular joint is technically not a part of the knee joint. The authors can emphasize the joints/articulations of the knee joint complex in this statement

Lines 37-42: this not a paragraph, but rather a run-on sentence that can be combined with the previous paragraph

Figure 3. the last sentence mentions axial plane. Do the authors mean the transverse plane corresponding to the longitudinal axis?

Lines 156-164: the authors mention internal-external and varus-valgus testing at various knee flexion angles. What were the angles that the knee was flexed during these trials. What was the rate of rotation to achieve the 5Nm torque? Were the tests performed in the same order of knee flexion for each varus-valgus and internal-external rotation? Did the authors attempt to reduce the influence of an order-effect?

Author Response

Dear Reviewer,

Thank you for your constructive feedback and insightful comments on our manuscript. We appreciate the time and effort you have taken to review our work. Below, we address each of your comments:

  1. Effectiveness of TKA Implants: We are pleased that you found our assessment of the TKA implants comprehensive and informative. We concur with your suggestion regarding the assessment of linear forces to determine stress distribution within the knee joint. In future studies, we will consider incorporating this aspect to provide a more holistic understanding of the biomechanics involved.

  2. Emphasis on Native vs. TKA Conditions: We acknowledge your observation regarding the emphasis on TKA testing. In our revised manuscript, we have made efforts to re-emphasize the comparison between the native knee and post-TKA conditions, ensuring that the distinctions and comparisons are clear to the reader, such e.g. in the beginning of the rersults section:
    "The clamped specimen knee joint was tested with the robot and provided motion and force-torque data first in native condition and later in the specific TKA configuration."

  3. Knee Joint Pivoting: You rightly pointed out the screw-home mechanism observed during knee flexion. We have examined the implants used in our study and can confirm that they do allow for such deviations, enabling the tibia to internally and externally rotate during knee flexion-extension motions. This information has been added to the manuscript for clarity, e.g. section 2.2: "... All implants allowed for internal-external rotation during knee flexion-extension motions."

  4. Proximal Tibiofibular Joint: We appreciate your attention to detail. We have revised the mentioned lines to emphasize the primary articulations of the knee joint complex, omitting the reference to the proximal tibiofibular joint and expressed the joint articulations:
    "The knee joint complex primarily comprises the medial tibiofemoral, lateral tibiofemoral, and patellofemoral articulations. These articulations collectively facilitate six degrees of freedom (DOFs), encompassing rotational motions such as flexion-extension, internal-external, and varus-valgus, as well as translational movements including anterior-posterior, medial-lateral, and compression-distraction"

  5. Paragraph Structure: Your observation regarding the run-on sentence in lines 37-42 is noted. We have restructured this section, combining it with the previous paragraph for better flow and coherence.

  6. Figure 3 - Axial Plane: Thank you for pointing this out. We intended to refer to the transverse plane corresponding to the longitudinal axis. The necessary correction has been made in the manuscript, further the view direction has been marked by an arrow in Figure 3 (a):
    "(a) 3D Slicer segmentation of the cadaveric knee joint, demonstrating the tibia, fibula, patella, and femur, along with the femoral (upper blue spheres) and tibial (lower blue spheres) trackers. (b) Tailored femoral component designs modeled on Triathlon CR (GmbH, Duisburg, Germany) implants, resized and rotated to -5° (internal), 0° (neutral), and +5° (external). The femoral components are displayed on the transverse (axial) plane, viewed distally, as indicated by the direction of the black arrow in (a)."

  7. Details on Testing Angles and Procedures: We apologize for any oversight in providing these details. In response to your queries:

    • The knee was flexed at angles of 0°, 30°, 60°, and 90° during the trials.
    • The rate of rotation to achieve the 5Nm torque was consistent at 0.5°/s.
    • Tests were performed in a consistent order of knee flexion for each varus-valgus and internal-external rotation to maintain uniformity.
    • We ensured that the tests were randomized to minimize any order-effect, and this has been clarified in the revised manuscript, in the section: 2.4:

      "

      A comprehensive biomechanical assessment of the cadaveric knee joint was conducted using the robotic testbench system. The evaluation involved varus-valgus and internal-external loading tests at multiple flexion angles, specifically at 0°, 30°, 60°, and 90°, for both the native knee joint and TKA conditions with varying implantation rotations.
      The rate of rotation to achieve the 5Nm torque was maintained consistently at 0.5°/s. To ensure uniformity and consistency in the testing procedure, tests were performed in a specific order of knee flexion for each varus-valgus and internal-external rotation. Furthermore, to minimize any potential order-effect, the sequence of tests was randomized.

      Force-torque data, position, and quaternion information were meticulously collected throughout the experiment. This data collection facilitated an in-depth analysis of the knee joint's biomechanical response under diverse loading conditions and allowed for robust comparisons between the native knee joint and the various TKA conditions. Tests were executed along the defined axis, up to a maximum load of 5 Nm. Each specific loading and flexion condition was repeated three times independently, with results captured at a 2500 Hz sampling rate. From these three independent tests, mean values and standard deviations were computed to provide a comprehensive overview of the biomechanical behavior of the knee joint under the tested conditions."

We hope that our responses address your concerns adequately. We are grateful for your valuable suggestions, which have undoubtedly enhanced the quality of our manuscript. We look forward to any further feedback you may have.

Best regards,

Nikolas Wilhelm